# Integrated Hypoxia Signaling and Oxidative Stress in Developmental Neurotoxicity of Benzo[a]Pyrene in Zebrafish Embryos

**DOI:** 10.3390/antiox9080731

**Published:** 2020-08-11

**Authors:** Yi-Chen Lin, Chang-Yi Wu, Chin-Hwa Hu, Tun-Wen Pai, Yet-Ran Chen, Wen-Der Wang

**Affiliations:** 1Department of Bioagricultural Sciences, National Chiayi University, Chiayi City 60004, Taiwan; lintina0212@gmail.com; 2Department of Biological Sciences, National Sun Yat-sen University, Kaohsiung 80424, Taiwan; cywu@mail.nsysu.edu.tw; 3Department of Bioscience and Biotechnology, National Taiwan Ocean University, Keelung 20224, Taiwan; chhu@mail.ntou.edu.tw; 4Department of Computer Science and Engineering, National Taiwan Ocean University, Keelung 20224, Taiwan; twp@csie.ntut.edu.tw; 5Department of Computer Science and Information Engineering, National Taipei University of Technology, Taipei 10608, Taiwan; 6Agricultural Biotechnology Research Center, Academia Sinica, Taipei 11529, Taiwan; yetran@gate.sinica.edu.tw

**Keywords:** benzo[a]pyrene, neurotoxicity, oxidative stress, hypoxia-inducible factor

## Abstract

Benzo[a]pyrene (B[a]P) is a polycyclic aromatic hydrocarbon formed by the incomplete combustion of organic matter. Environmental B[a]P contamination poses a serious health risk to many organisms because the pollutant may negatively affect many physiological systems. As such, chronic exposure to B[a]P is known to lead to locomotor dysfunction and neurodegeneration in several organisms. In this study, we used the zebrafish model to delineate the acute toxic effects of B[a]P on the developing nervous system. We found that embryonic exposure of B[a]P downregulates *shh* and *isl1*, causing morphological hypoplasia in the telencephalon, ventral thalamus, hypothalamus, epiphysis and posterior commissure. Moreover, hypoxia-inducible factors (*hif1a* and *hif2a*) are repressed upon embryonic exposure of B[a]P, leading to reduced expression of the Hif-target genes, *epo* and *survivin*, which are associated with neural differentiation and maintenance. During normal embryogenesis, low-level oxidative stress regulates neuronal development and function. However, our experiments revealed that embryonic oxidative stress is greatly increased in B[a]*P-*treated embryos. The expression of *catalase* was decreased and *sod1* expression increased in B[a]*P-*treated embryos. These transcriptional changes were coincident with increased embryonic levels of H_2_O_2_ and malondialdehyde, with the levels in B[a]*P-*treated fish similar to those in embryos treated with 120-μM H_2_O_2_. Together, our data suggest that reduced Hif signaling and increased oxidative stress are involved in B[a]*P-*induced acute neurotoxicity during embryogenesis.

## 1. Introduction

The nervous system is the most complex and important signal transmission system in the animal body, playing important roles in controlling nearly every aspect of an animal’s physiology and response to the environment. As such, the nervous system is centrally involved in signal transmission, automatic activities, cognition, memory, emotion and functional coordination of physiological systems. Development of the vertebrate nervous system begins with the process of neural induction, which refers to the specification of neuroectodermal cells that give rise to the neural plate. This process is regulated by complex interactions between Bmp, Wnt and Fgf signaling [1,2]. After neuroectodermal cells form the neural plate on the dorsal side of the embryo, the tissue is then remodeled and curls to form a neural tube along the central axis on the dorsal side of the embryo. Cells within the neural tube further differentiate to become mature neurons. Anomalies in the development of the nervous system can cause congenital mental retardation, neural tube defects, epilepsy, dementia and neurodegenerative disease; some such anomalies are caused by environmental chemical toxicants.

One major class of chemical toxicants is polycyclic aromatic hydrocarbons (PAHs), a group of semivolatile organic compounds that are ubiquitous in the environment. The widespread environmental distribution of these toxicants is because of their generation from the incomplete combustion of natural or artificial organic matter and anthropogenic plastic and petrochemicals [3,4]. Many PAHs and their epoxides are known to be highly cytotoxic, genotoxic (mutagenic and/or carcinogenic) and embryotoxic to vertebrates. Benzo[a]pyrene (B[a]P) is a five-ring PAH that is created by the incomplete combustion of organic matter; notably, it is found in cigarette smoke and grilled and broiled foods [5]. This molecule is classified as a Group 1 carcinogen by the International Agency for Research on Cancer (IARC), meaning that it is a known human carcinogen, and it can induce cancer in multiple organs of laboratory animals [6,7]. There is no known commercial or industrial use for B[a]P, and it is only purposefully produced for use in research. B[a]P is taken up by organisms from the air, water or diet [8,9]. Once taken up into cells, B[a]P induces the expression of Cytochrome p450 (CYP) monooxygenase enzymes through AhR/ARNT signaling, enhancing the catalysis of CYP-mediated metabolic reactions [10]. Research has shown that B[a]P exhibits carcinogenicity, teratogenicity, neurotoxicity and immunotoxicity in various species of experimental animals [11,12,13,14].

Many toxicants are known to damage cells by elevating the intracellular reactive oxygen species and the induction of oxidative stress has been implicated in various conditions of the nervous system, including neurodegenerative disease, attention deficit hyperactivity disorder, autism and dyslexia. Oxidative stress refers to elevated intracellular levels of reactive oxygen species (ROS) and is defined as an imbalance between free radicals and antioxidants in the cell or body [15]. In some normal physiological states, oxidative stress is known to regulate gene expression, cellular differentiation and aging-related mechanisms, among other processes [16,17,18]. On the other hand, oxidative stress that is induced by cellular exposure to toxicants often leads to cell and tissue damage [19,20]. ROS are derived from several intracellular sources, including mitochondria, NAD(P)H oxidase, xanthine oxidase and uncoupled nitric oxide synthase [21]. During embryogenesis, experimental enhancement or suppression of ROS alters embryonic development and growth, indicating a certain level ROS is required to regulate the normal progression of embryonic development [22]. Although excessive ROS has been implicated in the progression of Alzheimer’s disease (AD), Parkinson’s disease (PD) and other neurodegenerative diseases [23], it also plays an important role in the regulation of embryonic neurogenesis [24] and the dynamic control of ROS levels during embryogenesis is a key factor in neural development.

Hypoxia-inducible factors (Hifs), including Hif1, Hif2 and Hif3, function in oxygen homeostasis by regulating the genes responsible for glucose metabolism, erythropoiesis, angiogenesis, apoptosis and cell proliferation [25]. As the cell number in developing embryos increases, endogenous embryonic hypoxia stimulates the expression of Hif genes and stabilizes Hif proteins, which induce the expression of vascular endothelial growth factor (*vegf*) and *erythropoietin* (*epo*) to, respectively promote angiogenesis and erythrogenesis [26,27]. Additionally, Hif2a has been reported to maintain neural progenitor cell populations and promote neural differentiation by regulating survival during embryogenesis [28].

In the current study, we investigated the role of Hif signaling and oxidative stress in the neuropathology of B[a]P in developing zebrafish embryos by analyzing the expression of genes related to Hif signaling, examining the markers of oxidative stress. Embryonic exposure to B[a]P resulted in developmental hypoplasia of the brain, trigeminal ganglion cells and Rohon–Beard cells at 24 hpf. Moreover, reduced expression of *hif1a* and *hif2a* was coincident with the low expression of *survivin* and *epo* in B[a]*P-*treated embryos. Expression of genes that modulate oxidative stress and the embryonic level of H_2_O_2_ were also altered in B[a]*P-*treated embryos. These results provide novel insights into how neurological disorders may be caused by B[a]P, further delineating the molecular neuropathology of B[a]P during zebrafish embryogenesis.

## 2. Materials and Methods

### 2.1. Chemicals

B[a]P (No. B-1760) and other chemicals were purchased from Sigma-Aldrich (St. Louis, MO, USA). To eliminate the potential interference of chemical contaminants, all chemicals used in this study were of a molecular biology grade.

### 2.2. Animal Care and Maintenance

Wild-type (AB) and transgenic (Tg(elavl3:EGFP)) zebrafish were maintained in a recirculating housing system under standard laboratory conditions: 28.5 °C, pH 7–8, 14:10 h light:dark cycle. Fish were fed with commercial tropical zebrafish feed (Azoo, Taiwan) twice daily and live brine shrimp (Ocean Star International, USA) once daily. The two zebrafish strains were used in this study, AB and Tg(elavl3:EGFP), were both originally obtained from the Taiwan Zebrafish Core Facility at Academia Sinica, Taiwan and were maintained in the laboratory for more than 10 generations. Embryos were obtained through natural mating. Experimental animals were collected at the 1- to 4-cell stages and cultured in sea salt egg water (0.0375% sea salt in deionized distilled water). All experiments involving animals were conducted in accordance with the Institutional Animal Care and Use Committee (IACUC) at National Chiayi University, which approved the animal experiments and standard operating protocols (Approval No. 104043).

### 2.3. Benzo[a]pyrene Treatment and Embryo Processing

To determine the acute toxicity of ethanol on the developing neural system, 1- to 4-cell stage embryos were transferred to a petri-dish containing sea salt egg water. Eighty embryos by 4 cells stages were used in each group to increase throughput and reduce variability. Embryos were allowed to habituate in B[a]P solutions at the following concentrations: 0 μM (control, 0.1% DMSO), 10 μM and 20 μM. To aid visualization of zebrafish development, 0.2-mM 1-phenyl-2-thiourea (Sigma-Aldrich) was used in some experiments to inhibit pigmentation at 20–22 h post-fertilization (hpf); embryos were staged as described previously [29]. The Tg(elavl3:EGFP) transgenic embryos were treated with B[a]P at concentrations of 0 μM (control, 0.1% DMSO), 10 μM and 20 μM and their neural development was observed under fluorescence microscopy at 24 hpf. All experiments were performed with at least three independent replicates.

### 2.4. In Situ Hybridization Staining

Zebrafish embryos were exposed to 0 μM (control, 0.1% DMSO), 10 μM and 20 μM of B[a]P since 2 hpf. Embryonic exposure to B[a]P was terminated and fixed at 12, 24 and 48 hpf in 4% paraformaldehyde in phosphate-buffered saline (PBS) at pH 7.2 and processed using standard protocols [30]. Antisense probes labeled with digoxigenin-UTP (Roche) were synthesized by in vitro transcription using cDNA templates encoding *elavl3* [31], *shh* [32], *epo* [33] and *survivin* (*birc5a*) [34], as previously described. In situ hybridization staining was developed with NBT/BCIP substrate. Sense riboprobes served as the control for each antisense probe, all in situ staining results are shown in the Appendix A. All imaging was performed using an Olympus BX53 upright microscope equipped with an Olympus DP72 Digital Color Microscope Camera. All images were processed for contrast, brightness and color using Photoshop CS3 Extended Version 10.0 (Adobe; San Jose, CA, USA).

### 2.5. Embryonic MDA and Hydrogen Peroxide Measurement

The lipid peroxidation product, malondialdehyde (MDA), is a well-established marker of oxidative stress in cells and tissues. The level of MDA was quantified spectrophotometrically at 535 nm on an Epoch spectrophotometer (BioTek). A calibration curve for MDA in the embryos was prepared by treating the embryos with H_2_O_2_ at 15, 30, 60, 120 and 180 μM. Therefore, the MDA levels were calculated as equivalents to H_2_O_2_ exposure concentrations (μM), based on a comparison of signal to the calibration curves. Values were then compared with negative controls and expressed as a fractional increase in MDA level compared with negative control. Embryonic H_2_O_2_ measurements were made using the ferrous ion oxidation xylenol orange (FOX) assay [35]. Zebrafish embryos were homogenized in 0.5 mL ice-cold 80% methanol and centrifuged, which was followed by the collection of supernatants. The supernatants of samples and FOX reagent (100-μM xylenol orange, 250-μM ferrous ammonium sulfate, 100-mM sorbitol and 25-mM H_2_SO_4_ in methanol) were mixed in a 1:10 ratio, and the mixtures were incubated at room temperature for 30 min in the dark. The absorbance of the mixtures was measured at 560 nm, and the concentration of embryonic H_2_O_2_ was calculated using a standard curve (0, 20, 40, 60, 80 and 100 μM).

### 2.6. Real-Time Quantitative PCR

Total RNA was extracted using Trizol reagent (), and 3 μg of total RNA was reverse transcribed with MMLV high performance reverse transcriptase (Epicenter) for cDNA synthesis. Quantitative PCR reactions were performed with EvaGreen dye (Biotium) for low background and high resolution data. Quantitative RT–PCR analysis was performed, and signals were detected using the Rotor-Gene Q System (QIAGEN). Expression levels for each individual gene were normalized to the *actin* gene expression from the same sample. Results were analyzed by the 2^(−△△CT)^ formula, as previously described (Livak and Schmittgen, 2001). The following primers were used. *actin*: 5′-CACCTTGCAGCAGATGTGGA-3′, 5′-AAAAGCCATGCCAATGTTGTC-3′; *birc5a*: 5′-CACTCCAGAAAACATGGCTAAA-3′, 5′-CCATCCTTCCAGCTCTTT CA-3′; epo: 5′-TGAAGTCTGGGAAGCG ATG-3′, 5′-GCATGTGTAAGCCTGAC TGG-3′; *hif1a*: 5′-CCTTCTGCCACTCACTGTGT-3′, 5′-CGAGGAGGGTAAGG GTTGGA-3′; *hif2a*: 5′-AGCTCGACTTGCTCTTGCTA-3′, 5′-CTGACAAGCT ACTGCTGAGTGA-3′; cat: 5′-AGGGCAACTGGGATCTTACA-3′, 5′-TTTATG GGACCAGACCTTGG-3′; *sod1* (Cu-SOD): 5′-GTCGTCTGGCTTGTGGAGTG-3′, 5′-TGTCAGCGGGCTAGT GCTT-3′; *survivin* (*bir5b*): 5′-AGACGTTGCCTGCTGTTTCT-3′, 5′-TAGGCAATCTTCCGATGACC-3′, *epo*: 5′-TGATGCTGATGGT GCTGGAG-3′, 5′-GACTGGACCTCCTGAGCTTG-3′.

### 2.7. Statistical Tests

The toxicity of B[a]P on zebrafish developing neural cells was determined based on the number of neurons. All experiments were conducted in triplicate and the data were presented as the mean ± standard deviation. Significant differences in developing neural number between the treatment groups were evaluated using on-way ANOVA, followed by Tukey’s post hoc test. Values were considered statistically significant when *p* < 0.05 and highly significant when *p* < 0.01. All statistical analyses were performed using SPSS 21.0 (IBM).

## 3. Results

To study the toxic effects of B[a]P on the embryonic neural development of zebrafish, Tg(elavl3:EGFP) embryos were used. In these transgenic fish, GFP is specifically expressed in neurons. We exposed Tg(elavl3:EGFP) embryos to 0.1% DMSO (control), 10 μM and 20 μM B[a]P from 2 hpf until 24 hpf. Images of B[a]*P-*treated live embryos at 24 hpf did not show obvious morphological differences from the controls (Figure 1A–C). Neural development was then observed under fluorescent microscopy (Figure 1D–O). From the live images, we noticed that the fluorescence intensity in B[a]*P-*treated embryos was weaker than that in the control embryos. The number of trigeminal ganglion cells in the head and Rohon–Beard cells in the front-trunk, mid-trunk and tail regions were counted. In the head region, the numbers of trigeminal ganglion cells were 15 ± 1.2 in the controls, 9.3 ± 2.2 in 10 μM B[a]P and 6.5 ± 1.3 in 20 μM B[a]*P-*treated embryos (Figure 1P). There were 14.7 ± 1.7, 27.8 ± 2.1 and 41.3 ± 0.7 Rohon–Beard cells in front-trunk, mid-trunk and tail region of control embryos. These numbers were reduced to 9.2 ± 2.1, 23 ± 0.6, 27.1 ± 1.6 in front-trunk, mid-trunk and tail region of 10 μM B[a]P, and the values were 6.8 ± 0.7, 18.7 ± 1.6 and 19.3 ± 2.2 in front-trunk, mid-trunk and tail region of 20 μM B[a]P treated embryos, respectively. These results indicated that the number of developing trigeminal ganglion cells and Rohon–Beard cells was significantly reduced in B[a]*P-*treated embryos. We further examined the development of the central nervous system in the control and B[a]*P-*treated embryos by in situ hybridization staining with a riboprobe against *elavl3* mRNA, which is specifically expressed in telencephalon, midbrain, midbrain hindbrain boundary, midbrain hindbrain plate, hypothalamus and cranial ganglion (Figure 2A). Compared with the control embryos, the expression pattern of *elval3* was obviously altered in B[a]*P-*treated embryos (Figure 2A–C). Compared with control embryos, the expression intensity of *elavl3* in the telencephalon fore-midbrain region (Figure 2D–F, rectangular frame), hindbrain and cranial ganglion (Figure 2G–I) in B[a]*P-*treated embryos was weaker than that in the control embryos and the expression width in the hindbrain was increased in B[a]*P-*treated embryos (Figure 2D–F). Real time quantitative PCR assay indicated that the expression of *elavl3* was does-dependently reduced in B[a]P treated embryos (Figure 2J). Taken together, our results clearly indicate that B[a]P treatment induced embryonic neurodevelopmental toxicity.

### 3.1. Exposure of B[a]P Causes Reduced Expression of shh and islt1 in the Developing Neural System at 24 hpf

To explore the toxic mechanism of B[a]P on the developing embryonic nervous system, we first probed the expression of *sonic hedgehog* (*shh*), a signal for neurogenic induction, proliferation and neuroprotection in various cell types and *islet 1* (*isl1*), which encodes a LIM homeodomain transcription that plays an essential role in embryogenesis. The exposure of zebrafish to B[a]P led to reduced expression of *shh* in mid-dicencephalic organizer, basal plate and hypothalamus (Figure 3A–C). From the lateral view, we also noticed that the expression of *shh* in the notochord and neural tube floor plate was more curved in 20 μM B[a]*P-*treated embryos (Figure 3C). From the dorsal view, expression of *shh* was obviously reduced in the brain and notochord of B[a]*P-*treated embryos (Figure 3E–F). In untreated wild-type embryos, *isl1* was specifically expressed in the telencephalon, ventral thalamus, hypothalamus, epiphysis and posterior commissure at 24 hpf. Its expression was significantly reduced by B[a]P treatment in the telencephalon, ventral thalamus and hypothalamus (Figure 3H–J). Surprisingly, expression of *shh* was expanded in ventral thalamus and dramatically reduced in the hypothalamus, epiphysis and posterior commissure of 20-μM B[a]*P-*treated embryos (Figure 3J). The frontal view also revealed a low expression of *isl1* in telencephalon, ventral thalamus and hypothalamus (Figure 3K–M). Real-time quantitative PCR was further performed to confirm the expression reduction of *shh* and *isl1* in B[a]P treated embryos (Figure 3N,O) These results serve to define the specific brain regions affected by B[a]*P-*induced disruption of neural development.

### 3.2. No Obvious Neural Defects Are Present in B[a]P-Exposed Embryos at 12 hpf

To determine the embryonic timing at which B[a]P begins to affect the developing neural system, we performed in situ hybridization staining with an *elavl3* riboprobe in embryos at 12 hpf. At this time point, *elavl3* is expressed in cranial trigeminal ganglions cells, primary motor neurons and Rohon–Beard cells (Figure 4A,D,G). Based on the staining results, no obvious effects of B[a]P exposure could be observed on trigeminal cranial ganglion cells in the head region (Figure 4A–C) or in primary motor neurons and Rohon–Beard cells (Figure 4D–I). These observations suggest that the neurotoxic effects of B[a]P are not present at 12 hpf, suggesting that the window of vulnerability begins at later stages.

### 3.3. B[a]P Exposure Alters the Expression of Hypoxia-Related and Oxidative Stress-Related Genes

Many studies have shown that the expression of Hifs and oxidative stress-related genes occurs in organisms exposed to chemical stresses [36], and the expression of these genes is also closely associated with embryonic neural development [37]. To assess whether the expression of these genes is altered by B[a]P exposure in the developing neural system, quantitative RT-PCR was performed to measure the expression of *hif1a*, *hif2a*, *calatase* and *sod1* (encodes Cu-containing superoxide dismutase; CuSod). Interestingly, the levels of *hif1a*, *hif2a* and *calatase* were reduced (Figure 5A–C) and the level of *sod1* was increased (Figure 5D) in B[a]*P-*treated embryos compared with the controls. The low expression of *hif1a* and *hif2a* was unexpected, so we further examined the expression of *survivin*, a target of *hif2a* that is known to regulate embryonic neural development. In situ hybridization results indicated that *survivin* was specifically expressed in the brain and neural tube of wild-type embryos at 24 hpf (Figure 6A,D) and the expression level was obviously reduced in B[a]*P-*treated embryos (Figure 6B,C,E,F). The expression levels were quantified and the results confirmed that B[a]P was able to dose-dependently reduce the expression (Figure 6M). We also detected another Hif target gene, *epo* (Figure 6G–L), which is expressed in brain during embryogenesis and is functionally associated with neural differentiation and proliferation [38,39,40]. The quantitative results indicated that the expression of epo was also dose-dependently reduced by B[a]P treatment (Figure 6N), and the effect sizes at each dose were similar to those for *hif2a* in B[a]*P-*treated embryos.

### 3.4. Exposure to B[a]P Increases Hydrogen Peroxide and Lipid Peroxidation

In many biological systems, exposure to chemical toxins often results in the cellular stress associated with induction or repression of cellular reactive oxygen species (ROS), such as hydrogen peroxide [41,42,43] and lipid peroxidation [44,45]. A proper embryonic level of hydrogen peroxide plays an important role in controlling axon pathfinding during zebrafish development; however high levels of cellular H_2_O_2_ result in the impairment of cells and embryogenesis [37,46]. Moreover, the levels of ROS and lipid peroxidation are influenced by a variety of enzymes. For example, catalase functions to decompose H_2_O_2_ to water and oxygen [46] and SOD1 (CuSod) catalyzes the dismutation of superoxide to hydrogen peroxide and oxygen [47]. Since we saw effects on the expression of *catalase* and *sod1* in B[a]*P-*treated embryos (Figure 5C,D), we wondered whether cellular H_2_O_2_ and the lipid peroxidation product, MDA, may be affected in B[a]*P-*treated embryos. In the positive control group, we exposed zebrafish embryos to 120 μM H_2_O_2_ combined with 0.1% DMSO and the embryonic levels of H_2_O_2_ and lipid peroxidation in the embryos were detected with the FOX and MDA assays, respectively. Our results indicated that the embryonic levels of H_2_O_2_ and MDA were increased in B[a]*P-*treated embryos. Surprisingly, the levels of these ROS markers in 10 μM B[a]*P-*treated embryos were similar to the positive control embryos (Figure 7A) and the levels of both markers in 20-μM B[a]*P-*treated embryos were higher than those in positive control embryos. Thus, the altered expression of *catalase* and *sod1* were coincident with high levels of H_2_O_2_ (Figure 7A) and lipid peroxidation products (Figure 7B) in B[a]*P-*treated embryos. Moreover, B[a]*P-*induced embryonic oxidative stress is likely to at least partially mediate the toxic effects of B[a]P on developing neural system during embryogenesis.

## 4. Discussion and Conclusions

There is increasing awareness of the relationship between environmental pollutants, oxidative stress and neurodegenerative diseases. In this study, we used zebrafish embryos to investigate the toxic effects of B[a]P and explore the association between its neural toxicity, the Hif-pathway and oxidative stress. Zebrafish embryos were developmentally exposed to B[a]P at 10 and 20 μM. Developing neural cells were monitored in B[a]*P-*exposed Tg(elavl3:EGFP) transgenic embryos, revealing the reductions in trigeminal ganglion cells in the head region and Rohon–Beard cells in the trunk. We noticed that the reduced cell levels were more significant in head and front-trunk than in the mid-trunk and tail (Figure 1). We further examined the developing neural cells by in situ hybridization for *elavl3*, *shh* and *isl1*; these results also revealed clear developmental toxicity to neural cells and showed more significant effects in the head and front trunk than in the posterior regions (Figure 2 and Figure 3).

Our results also suggest that exposure to B[a]P at early embryonic stages results in the expression of neural regulatory genes in the head and front-trunk. In situ hybridization staining for *shh* indicated that its expression level and pattern was severely affected in the mid-diencephalic organizer, hypothalamus and basal plate brain regions, without obvious effects in the trunk regions (Figure 2A–G). During embryonic neurogenesis, *shh* is expressed in the developing hypothalamus and is essential for hypothalamic induction and mid-diencephalic organizer development [48,49,50]. Staining for *isl1* expression also clearly revealed the effects of B[a]P toxicity on brain development. At 24 hpf, *isl1* is normally specifically expressed in telencephalon, ventral thalamus, hypothalamus, epiphysis and posterior commissure regions of the developing brain (Figure 3H). However, in B[a]*P-*treated embryos, the expression areas and intensity of *isl1* in these regions were dramatically reduced (Figure 3I,J). To explore when B[a]P toxicity is first observed, we further examined neural development in B[a]*P-*treated embryos at an earlier time-point. No developmental toxicity was observed in B[a]*P-*treated embryos at 12 hpf (Figure 4). These results indicated that the neurotoxicity of B[a]P is likely to occur at embryonic stages later than12 hpf.

Many reports have shown the induction of oxidative stress upon exposure to environmental toxins [51] or hypoxia [52]. Indeed, oxidative stress is arguably the most common toxic mechanism for environmental agents [53,54]. Hifs belong to the basic helix-loop-helix-PER-ARNT-SIM (bHLH-PAS) subfamily of bHLH transcription factors and are often centrally involved in the cellular response to oxygen levels. Our results indicated that the expression of *hif1a* and *hif2a* was dose-dependently reduced in B[a]*P-*treated embryos, suggesting the Hif pathway may be involved in the neurodevelopmental toxicity of B[a]P. During embryogenesis, Survivin is especially important in neural progenitors of mammalian and zebrafish embryos [55]. During brain development, *survivin* is highly expressed in neural progenitor cells and plays an essential role in the maintenance of developing neural cells [56]. Antisense morpholino-induced knockdown of the zebrafish Hif2a protein was shown to cause a deficit in *survivin* transcription, which abrogates neural cell development [28]. This mechanism may explain our observation that embryonic exposure of B[a]P dose-dependently reduced the expression of *survivin* in concert with *hif1a* and *hif2a* (Figure 5A,B and Figure 6M).

Epo is widely known for its role in erythropoiesis during embryogenesis and in adults. Many reports have shown that Epo signaling also plays an important role in the neuroprotection of the adult brain [57,58] and neural development during embryogenesis [59,60]. During embryogenesis, the developing nervous system requires precise regulation of oxygen levels. Low levels of oxygen induce Hif gene expression, which enhances the expression of *epo* to regulate erythrogenesis [61]. Our results indicated that the expression levels of *hif1a* and *hif2a* were reduced in B[a]*P-*treated embryos. Consistently, the expression of *epo* also exhibited a dose-dependent reduction in B[a]*P-*treated embryos. These results suggest that embryonic exposure to B[a]P reduces Hif-mediated signaling, which may be responsible for the reduced expression of *survivin* and *epo*, resulting in defective neural development.

Inhibition of *hif1a* expression is known to enhance the generation of ROS and reduce the transcription of primary antioxidant enzymes, which can cause pathology in multiple organs [62]. Expression of ROS-related genes, *calatase* and *sod1*, was altered in B[a]*P-*treated embryos. Both Catalase and Sod1 participate in ROS detoxification. Sod1 is a cytosolic antioxidant enzyme that converts the highly reactive superoxide radical to H_2_O_2_, which can be eliminated by enzymes, such as Catalase. The induced expression of Sod1 and reduced expression of Catalase suggested that embryonic exposure to B[a]P may increase the cellular level of H_2_O_2_. As predicted, the embryonic H_2_O_2_ level was dose-dependently induced after B[a]P exposure. While high levels of H_2_O_2_ are widely known to cause cytotoxic effects, many recent reports have shown that the molecule is also an important regulator of eukaryotic signal transduction [63,64].

Surprisingly, the embryonic H_2_O_2_ level in 10 μM B[a]*P-*treated embryos was similar to the positive control embryos treated with 120 μM H_2_O_2_. During zebrafish embryogenesis, dynamic changes in endogenous embryonic H_2_O_2_ level may play an important role in regulating proper embryogenesis [65]. Excessive cellular H_2_O_2_ results in lipid peroxidation, triggers cell death [66,67,68] and induces neutrophil infiltration, inflammation by modulating the immune system [69,70]. Exposure to B[a]P is known to play a role in lung carcinogenesis induced by tissue inflammation [70,71]. In current years, the potential neurotoxicity of B[a]P inducing oxidative stress-mediated neurotoxicity and causing behavioral alterations and oxidative stress in animal models was reported [72,73,74]. B[a]P can reach the encephalic central nervous tissues by crossing the blood–brain barrier [75,76,77]; this leads to increase nervous oxidative stress in the nervous system which in turn resulting in behavioral changes and alterations to the expression of antioxidant enzymes (e.g., superoxide dismutase, catalase and glutathione peroxidase and cellular lipid peroxidation [77,78]. Epidemiological studies have demonstrated that exposure to B[a]P causes neurological abnormalities such as cognitive impairment, learning difficulties, parasympathetic dysregulation and short-term memory loss [79,80,81]. Embryonic B[a]P exposure increases the risk of neural hypoplasia. Neural tube defects are the most common and severe congenital malformations due to the disturbance of normal neural tube closure during early embryogenesis. Maternal B[a]P exposure increases fetal neural tube defects, neural apoptosis and the expression of the oxidative stress related genes, *cyp1a1*, *sod1* and *sod2*. Neural apoptosis and oxidative stress-related gene expression can be attenuated by ectopic supplementation of antioxidants, such as vitamin E or retinoic acid [82,83]. This shows that B[a]P exposure induces embryonic neural defects that may involve increased oxidative stress.

Based on our findings, we conclude that Hif signaling-mediated embryonic oxidative stress is likely to be involved in the developmental neurotoxicity of B[a]P and the tissue inflammation process. Therefore, our study further elucidates the molecular basis of B[a]P neurotoxicity during embryogenesis.

## Figures and Tables

**Figure 1 antioxidants-09-00731-f001:**
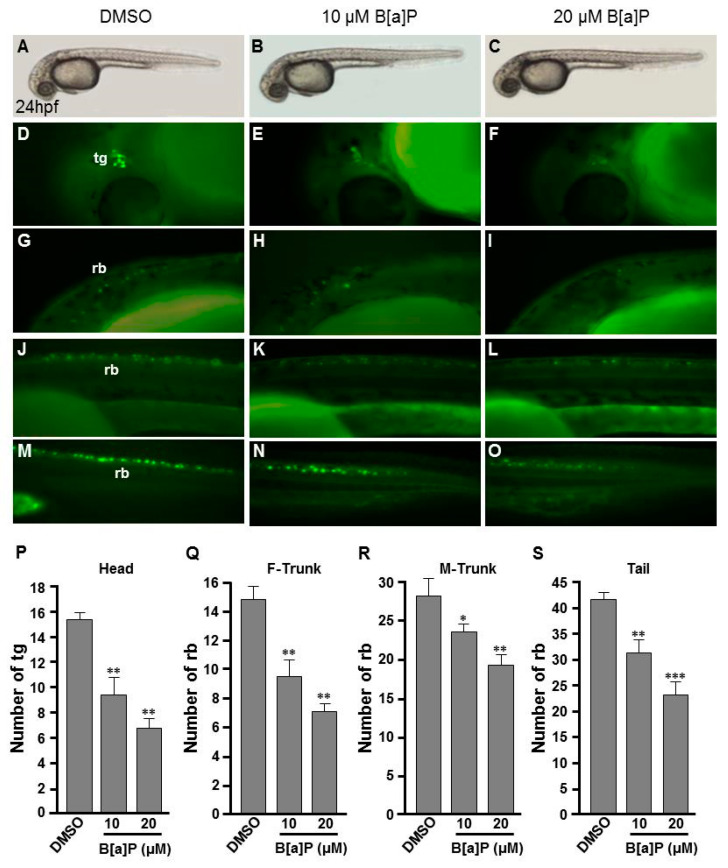
Developmental neurotoxicity of B[a]P revealed by the Tg(elavl3:EGFP) model. Tg(elavl3:EGFP) zebrafish embryos received the indicated treatments: 0 μM. (control, 01% DMSO; (**A**,**C**,**F**,**I**,**L**)), 10-μM B[a]P (**B**,**D**,**G**,**J**,**M**) and 20-μM B[a]P (**C**,**E**,**H**,**K**,**N**). (**A**–**C**) Lateral view of the live embryos at 24 hpf after treatment with 0.1% DMSO (control; **A**), 10-μM B[a]P (**B**) and 20-μM B[a]P (**C**); (**D**–**O**) Effects of B[a]P on neurogenesis using the Tg(elavl3:EGFP) model. The embryos treated with 0.1% DMSO (control; (**D**,**G**,**J**,**M**)), 10-μM B[a]P (**E**,**H**,**K**,**N**) and 20-μM B[a]P (**F**,**I**,**L**,**O**) were collected at 24 hpf. Representative fluorescent images are shown for embryonic trigeminal ganglions (tg) in the head region (**D**–**F**) and Rohon–Beard cell (rb) in the front trunk (**G**–**I**), mid-trunk (**J**–**L**) and in tail regions (**M**–**O**). (**P**–**S**) The numbers of tg cells in the head region and rb cells in front-trunk (F-trunk), mid-trunk (M-Trunk) and tail were counted from 10 embryos in each treatment group. Data are shown as mean ± standard deviation. * *p* < 0.05, ** *p* < 0.01, *** *p* < 0.001 compared with the control.

**Figure 2 antioxidants-09-00731-f002:**
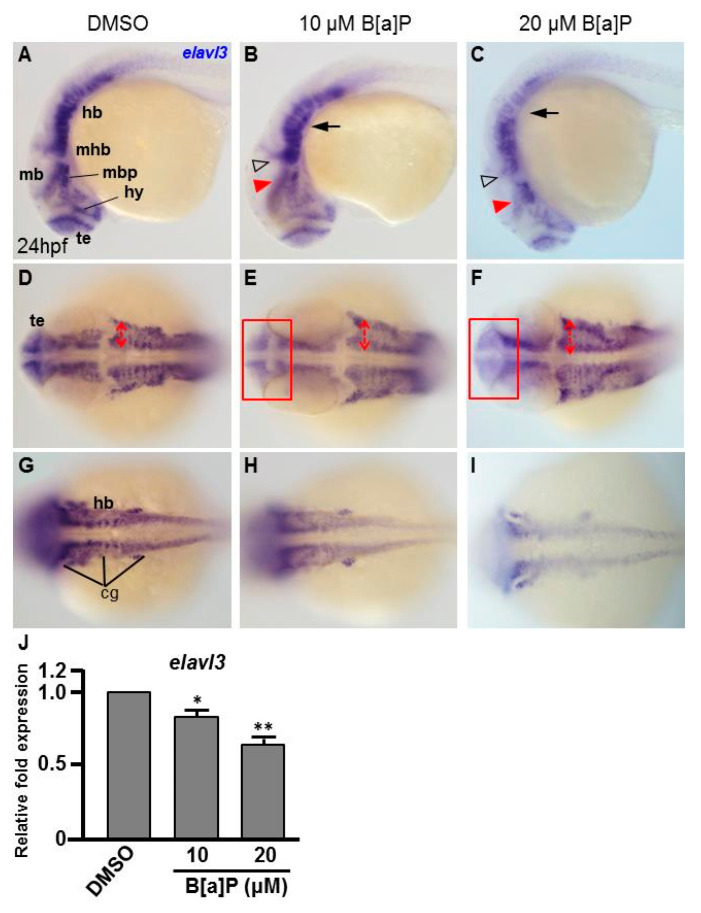
B[a]*P-*treated embryos exhibit defective nervous system development. In situ hybridization staining was performed with a riboprobe against *elavl3*, a marker gene specifically expressed in telencephalon (te), hypothalamus (hy), midbrain (mb), midbrain hindbrain boundary (mhb) and plate (mhp) and hindbrain (hb) of the developing brain (**A**). (**A**–**I**) Zebrafish embryos exposed to 0 μM. (control, 01% DMSO), 10-μM B[a]P and 20-μM B[a]P were analyzed at 24 hpf. Embryos in (**A–C**) are shown from the lateral view; the red arrowheads indicate developmental defects in the midbrain (mb), and the hollow arrowheads indicate developmental defects in midbrain-hindbrain boundary (mhb); (**D**–**F**) Embryos are shown from the lateral view. The developing telencephalon (te; rectangles in (**E**,**F**)) was wider and the expression of *elavl3* was weaker than in the control embryos; (**G**–**I**) Images are focused on the hindbrain (hb) region. The development of cranial ganglia (cg) was dose-dependently affected by B[a]P treatment; (**J**) Expression alteration of *elavl3 i*n B[a]P was quantified by real-time quantitative RT–PCR (* *p* < 0.05, ** *p* < 0.01 compared with the control).

**Figure 3 antioxidants-09-00731-f003:**
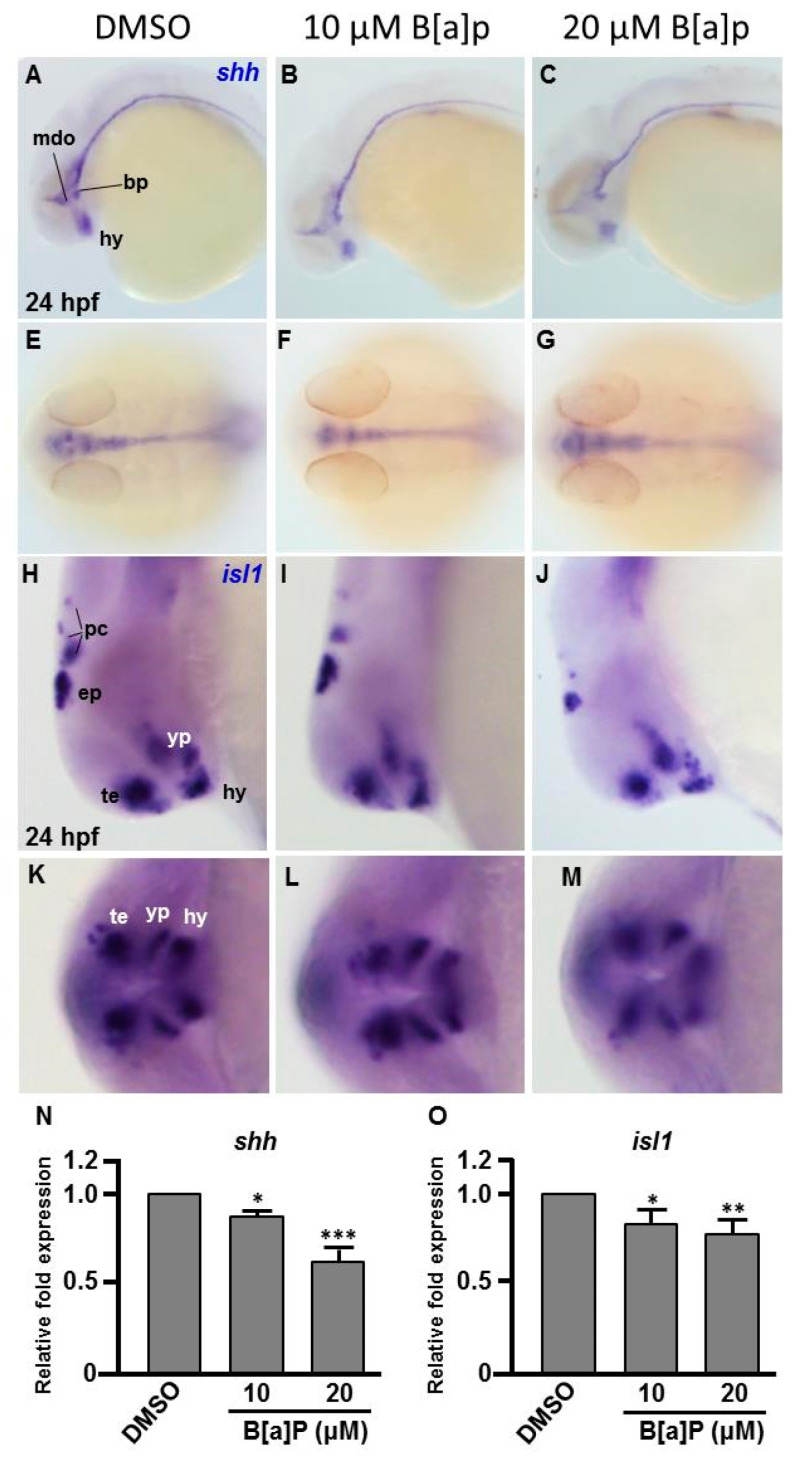
Expression patterns of *shh* and *isl1* are dose-dependently affected by B[a]P exposure. Wild-type (AB) zebrafish embryos were treated with 0 μM. (control, 01% DMSO), 10-μM B[a]P and 20-μM B[a]P. The embryos were fixed at 24 hpf. Expression of *shh* (**A**–**G**) and *isl1* (**H**–**M**) was analyzed by in situ hybridization staining. Images show the lateral view in (**A**–**C**,**H**–**J**). Images in (**E**–**G**) show the dorsal view. Images in (**K**–**M**) show the frontal view; (**B**,**C**) Expression of *shh* was specifically expressed in the mid-diencephalic organizer (mdo), hypothalamus (hy) and basal plate (bp) of B[a]*P-*treated embryos; (**E**–**G**) The expression pattern of *shh* in the brain and floor plate (fp) in 20-μM B[a]*P-*treated embryos; (**H**–**M**) The development of encephalic regions was examined by in situ hybridization staining by probing for the *islet1* gene (*isl1*) which is specifically expressed in telencephalon (te), ventral thalamus (vp), hypothalamus (hy), epiphysis (ep) and posterior commissure (pc) in embryos at 24 hpf. (N, **O**) Expression alteration of *shh* and *isl1* in B[a]P was quantified by real-time quantitative RT–PCR (* *p* < 0.05, ** *p* < 0.01, ****p* < 0.001 compared with the control).

**Figure 4 antioxidants-09-00731-f004:**
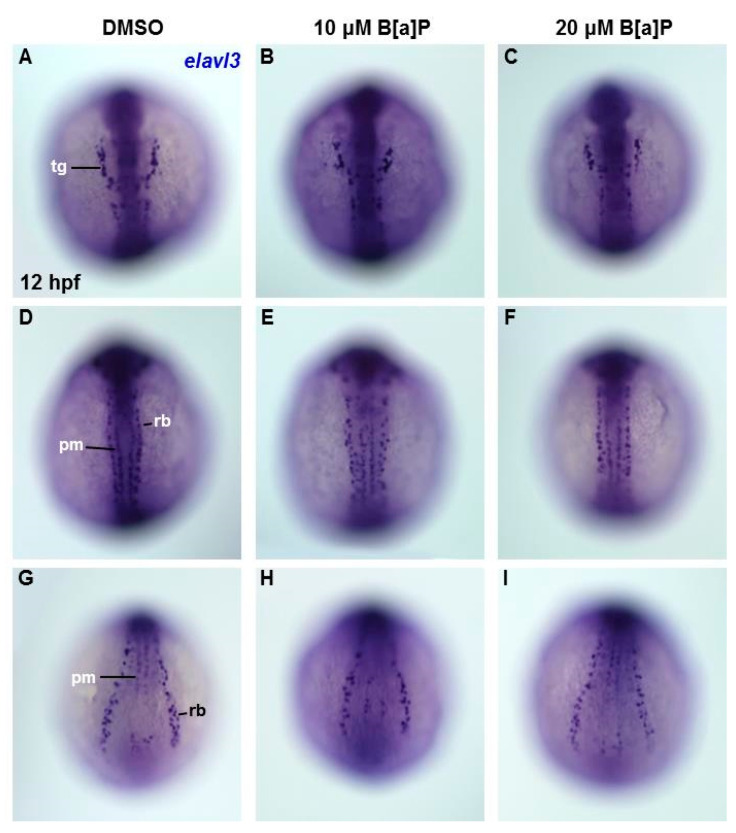
Lack of obvious neural cell defects in B[a]*P-*treated embryos at 12 hpf. In situ hybridization staining with a probe against *elavl3* was performed to analyze the early development of neural cells (**A**–**I**); all images show the dorsal view. Images in (**A**–**C**) are focused on the anterior trunk; (**D**–**F**) are focused on mid-trunk region and (**G**–**I**) are focused on the tail region. tg, trigeminal ganglion; pm, primary motor neurons; rb, Rohon–Beard cells.

**Figure 5 antioxidants-09-00731-f005:**
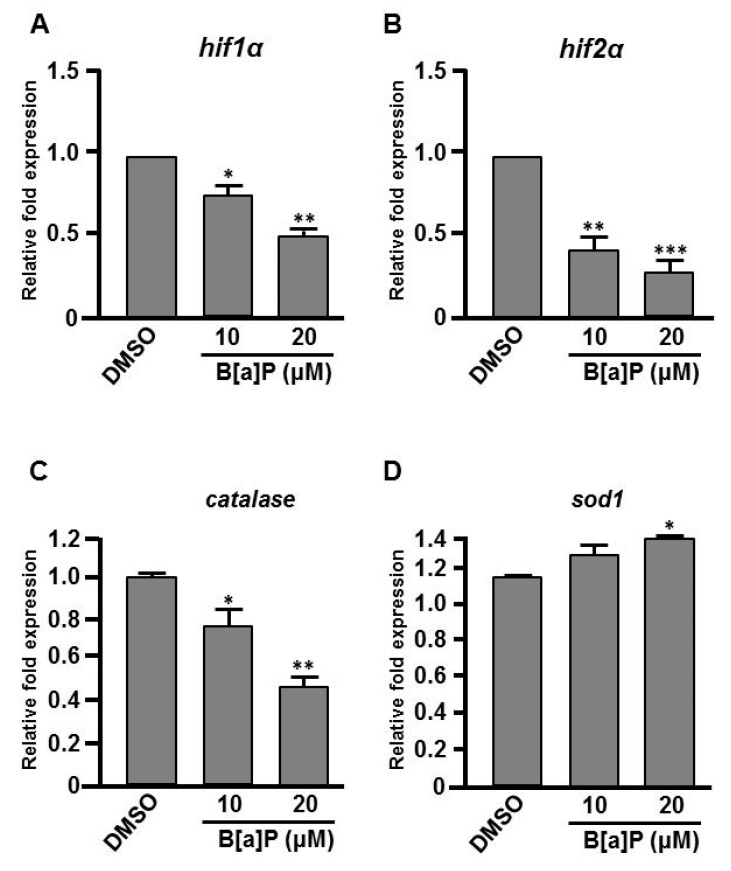
Expression levels of Hifs and oxidative stress-related genes are altered in B[a]P treated embryos. Zebrafish embryos were exposed to 0 μM. (control, 01% DMSO), 10 and 20 μM of B[a]P from 2 hpf until 24 hpf. Expression of genes encoding (**A**) Hif1α, (**B**) Hif2α, (**C**) Catalase and (**D**)Sod1 was examined by real-time quantitative RT–PCR. * *p* < 0.05, ** *p* < 0.01, *** *p* < 0.001 compared with the control.

**Figure 6 antioxidants-09-00731-f006:**
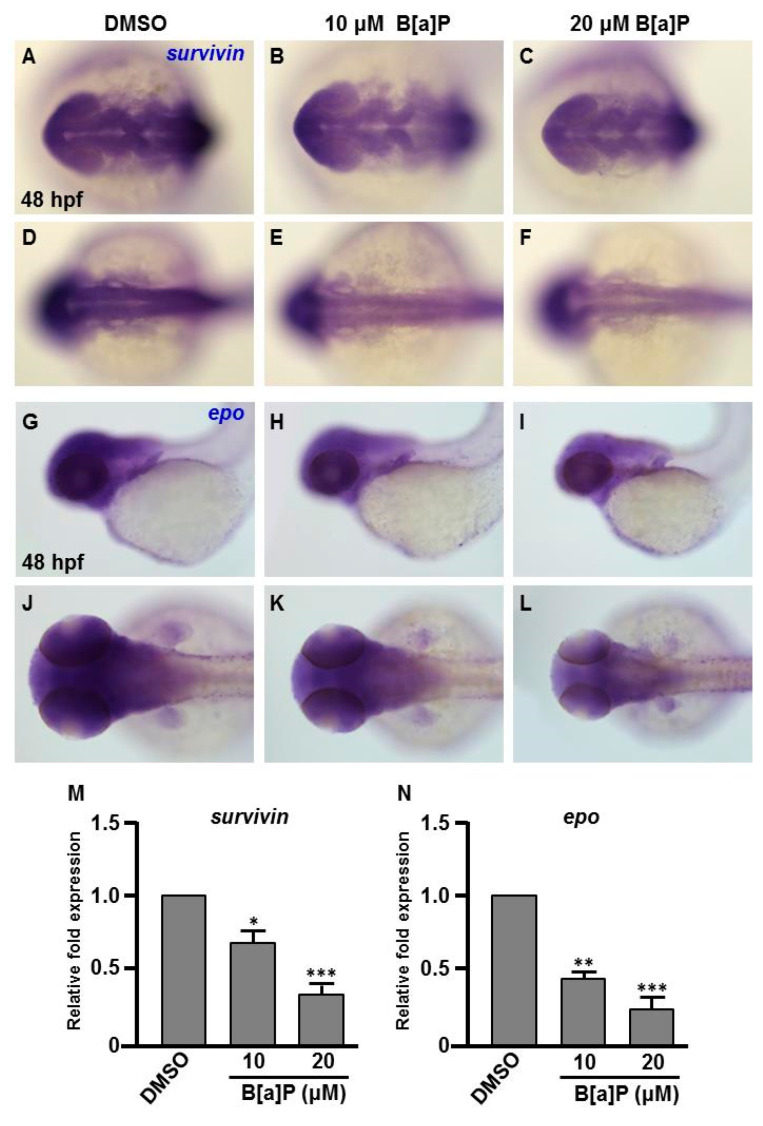
Expression levels of Survivin and Epo genes are reduced in B[a]P treated embryos. Zebrafish were exposed to 0 μM (control, 01% DMSO), 10 and 20 μM of B[a]P from 2 hpf until 24 hpf. Whole-mount in situ hybridization staining was performed with riboprobes against *survivin* (**A**–**F**) and *epo* (**G**–**L**) at 24 hpf. The staining levels of *survivin* (**M**) and *epo* (**N**) were quantified. * *p* < 0.05, ** *p* < 0.01, *** *p* < 0.001 compared to control.

**Figure 7 antioxidants-09-00731-f007:**
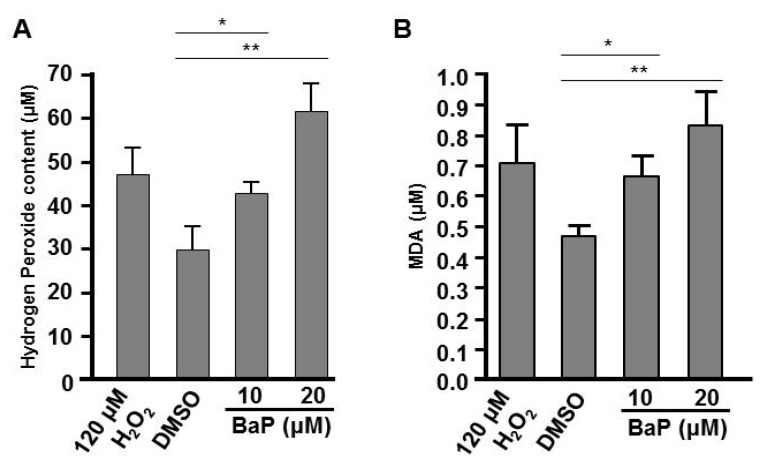
Exposure to B[a]P increases embryonic hydrogen peroxide and lipid peroxidation. (**A**) Embryonic hydrogen peroxide content was determined by the FOX assay and (**B**) the level of lipid peroxidation was examined by the malondialdehyde (MDA) assay. * *p* < 0.05, ** *p* < 0.01.

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
