# Peer review of "Integrated Hypoxia Signaling and Oxidative Stress in Developmental Neurotoxicity of Benzo[a]Pyrene in Zebrafish Embryos"

_antioxidants, 2020, doi:10.3390/antiox9080731_

Round 1

Reviewer 1 Report

An excellent paper on benzo[a]pyrene studying hypoxia and related oxidative stress in zebrafish embryos / Introductions is up to date, results are ok, and discussion is well presented.

One question (maybe this could be inserted in the methodology): the probes were measured in duplicates or just simple like that?

I recommend acceptance.

Author Response

Response to Reviewer 1 Comments

Point 1: One question (maybe this could be inserted in the methodology): the probes were measured in duplicates or just simple like that?

Response 1:

Thanks for review’s comment.

All the experiments were performed three independent replicates, and all the data from each independent experiment exhibited the consistence. All template plasmids which contain marker gene fragments were confirmed by DNA sequencing before they were used for riboprobe synthesis.

Dear Editor, Antioxidants journal

We would like to take this opportunity to thank the members of the editorial board and the reviewers for their helpful and constructive criticism on our manuscript “Integrated hypoxia signaling and oxidative stress in developmental neurotoxicity of Benzo[a]Pyrene in zebrafish embryos.” (ID: antioxidants-877402) as well as offering us the opportunity to submit a manuscript to the special issue. In this revised version, we have carried out the experiments that the reviewers suggested and revised the manuscript accordingly. Please find attached a point-by-point response to the reviewer’s concerns. The manuscript has been revised, using the “track changes” feature in Microsoft Word. We very much hope the revised manuscript is accepted for publication in the Antioxidants journal.

Sincerely yours,

Wen-Der Wang

National Chiayi University

[email protected]

Reviewer 2 Report

The work of  Lin et al. presents research regarding the neurotoxic effects of benzo[a]pyrene as assessed in zebrafish embryos. The authors found that benzo[a]pyrene induced morphological hypoplasia in the embryos and  gene expression analysis revealed the down-regulation of sonic hedgehog and islet 1. The authors determined the involvement of hypoxia signaling and oxidative stress induction in benzo[a]pyrene-induced neurotoxicity. Hypoxia-inducible factors hif1a and hif2a were found to be down-regulated along with target genes epo and survivin. Benzo[a]pyrene induced the levels of H2O2 and malondialdehyde and decreased the expression of catalase and increased sod1.

Overall, the manuscript is well-written and clearly presented, providing some new information to the current knowledge regarding benzo[a]pyrene-induced neurotoxicity. I have the following questions for the authors regarding their study:

Major points:

Could the authors explain the use of the selected concentrations of benzo[a]pyrene for their experiments. The selected concentrations seem quite high, especially for the  analysis of gene expression. Previous research of Knecht et al. (2017) regarding the use of zebrafish embryos  assessed the neurotoxic  effects of benzo[a]pyrene at the maximum concentration of 4 µM.

As mentioned by the authors the role of benzo[a]pyrene in the induction of neurotoxicity was previously examined. I think that the article would benefit from referencing and including some discussion regarding previous work related to the role of oxidative stress in benzo[a]pyrene-induced neurotoxicity. Some previous studies address some of the same objectives as the authors  and therefore could be discussed. For example Saunders et al. (2006; J Appl Toxicol 26 : 427-438) addressed the effects of benzo[a]pyrene on the activity of antioxidant enzymes (such as SOD and catalase) as well as levels of malondialdehyde in selected rodent brain regions. In another study, Dutta et al. (2010; PlosOne), examined the effects of benzo[a]pyrene on intracellular ROS induction and antioxidant proteins in neurons.

Minor points:

The y axis should be titled in Figures 5 and 6.

Author Response

Response to Reviewer 2 Comments

Point 1: Could the authors explain the use of the selected concentrations of benzo[a]pyrene for their experiments. The selected concentrations seem quite high, especially for the analysis of gene expression. Previous research of Knecht et al. (2017) regarding the use of zebrafish embryos assessed the neurotoxic effects of benzo[a]pyrene at the maximum concentration of 4 µM.

Response 1:

Thanks for reviewer’s comments.

In this study, the concentration of B[a]P was refer to the data that was published by Batel et al. (2018, published in Environmental Pollution journal), the zebrafish exposure concentration of B[a]P at 10 and 20 μM was used. In this manuscript, the acute neurotoxicity of B[a]P was observed and analyzed at 12, 24 and 48 hpf. The exposure time of zebrafish embryos in B[a]P in our study was shorter than that in Knecht’s study (exposure time from 2 hpf to 12, 24 or 48 hpf in our study and from 6 hpf to 120 hpf in Knecht’s study).

Point 2: The y axis should be titled in Figures 5 and 6.

Response 2:

Thanks for reviewer’s comment, and I am sorry for these omissions.

I have added the titles to the y axis in Figure 5 and 6.

Reviewer 3 Report

Dear Authors,

as requested, I reviewed the manuscript (ID diagnostics-877402) "Integrated hypoxia signaling and oxidative stress in developmental neurotoxicity of Benzo[a]Pyrene in zebrafish embryos", by Yi-Chen Lin, Chang-Yi Wu, Chin-Hwa Hu, Tun-Wen Pai, Yet-Ran Chen, and Wen-Der Wang.

The manuscript deals with the effects of Benzo[a]pyrene (B[a]P) as environmental contaminant potentially affecting the nervous system’s function. By using the zebrafish model, authors investigated some acute toxic effects of B[a]P on the developing nervous system. They described that embryonic exposure of B[a]P downregulates expression of genes such as shh and isl1, also suggesting some morphological alterations (telencephalon, ventral thalamus, hypothalamus, epiphysis and posterior commissure). Moreover, authors studied the expression of hypoxia-inducible factors (hif1a and hif2a), epo and surviving. Their experiments showed oxidative stress increasing in B[a]P-treated embryos. Together, data suggest the association of reduced Hif signaling and increased oxidative stress with B[a]P-induced acute neurotoxicity during embryogenesis.

Overall, the reviewer finds some merit in the novelty of the information, nevertheless the manuscript shows some serious issues, thus must be explicitly improved.

I made the following minor comments:

  • In section 2.3 and on, authors should unify the indication of concentrations, i.e. avoiding to use 0% (percent) and 10 or 20 µM (micromolar)
  • Overall, many statements along the manuscript are in an almost hasty form, so they sound not entirely correct; please review the manuscript keeping in mind the explicit meaning of any statement.

Major comments:

  • Lacking of sense probes as control for each antisense probe in situ assays is a serious shortcoming that needs to be amended
  • In section 2.4 as well as in the related results, exposure time and embryonic stages must be clearly stated
  • In describing results of in situ hybridizations, authors usually give “quantitative” hints without the support of actual quantitative evaluation. For instance, regarding the elavl3 in situ assays, they state “dose-dependently reduced” without giving numerical demonstration. Also, for shh authors describe “obvious” differences: here, as in other points, they use the terms obvious or obviously in a criticizable manner, since nothing is obvious based only on the subjectivity of qualitative observation, especially when authors want to describe a scientific evidence. These issues concerning the interpretations of the in situ assays should be amended by performing staining quantitation for all the in situ images of all the gene-specific assays, just like the authors did for the assays of survivin and epo…
  • Section 3.4: if the authors want to compare effects of B[a]P with a positive control such as H2O2, they should demonstrate or, alternatively, describe and add comments on the potentially different routes of exposure between the two ROS-inducing situations on embryonic cells…

In general, it is understandable that the authors demonstrated expertise in their research and in managing the data. Nevertheless, I state that the manuscript is not publishable in the present form.

Thank you very much for your attention to my opinion.

Author Response

Response to Reviewer 3 Comments

Point 1: In section 2.3 and on, authors should unify the indication of concentrations, i.e. avoiding to use 0% (percent) and 10 or 20 µM (micromolar)

Response 1:

Thanks the reviewer's for valuable comment.

In the manuscript, I have changed the 0.1 % DMSO (control) to 0 µM (control, 0.1 % DMSO).

Point 2: Lacking of sense probes as control for each antisense probe in situ assays is a serious shortcoming that needs to be amended.

Response 2:

Thanks for your comment.

In situ hybridization staining results with sense riboprobes were added to SFigure 1, and the description was added to section 2.4 (line 137). No background staining signal was presented in all embryos stained with sense riboprobes of each gene.

Point 3: In section 2.4 as well as in the related results, exposure time and embryonic stages must be clearly stated.

Response 3:

I have added the exposure time and the embryonic stages to the section 2.4.

Point 4: In describing results of in situ hybridizations, authors usually give “quantitative” hints without the support of actual quantitative evaluation. For instance, regarding the elavl3 in situ assays, they state “dose-dependently reduced” without giving numerical demonstration. Also, for shh authors describe “obvious” differences: here, as in other points, they use the terms obvious or obviously in a criticizable manner, since nothing is obvious based only on the subjectivity of qualitative observation, especially when authors want to describe a scientific evidence. These issues concerning the interpretations of the in situ assays should be amended by performing staining quantitation for all the in situ images of all the gene-specific assays, just like the authors did for the assays of survivin and epo….

Response:

Many thanks for reviewer’s comment and suggestion.

I have quantified the expression alteration of elval3, isl1 and shh by real-time quantitative PCR, the results were added Figure 2, 3 and also added the descriptions to the manuscript (line 210-211 and line 296-297).

Point 5: Section 3.4: if the authors want to compare effects of B[a]P with a positive control such as H2O2, they should demonstrate or, alternatively, describe and add comments on the potentially different routes of exposure between the two ROS-inducing situations on embryonic cells…

Response 5:

Thanks for reviewer’s comment.

I have added a brief description to the manuscript. In order to compare the embryonic level of hydrogen peroxide and lipid peroxidation between hydrogen peroxide-treated and B[a]P-treated embryos, the 120 μM H2O2 was prepared with 0.1% DMSO containing-sea salt egg water.

Round 2

Reviewer 2 Report

Following the authors’ response to my review, I feel that my questions were not fully answered. The authors addressed my inquiry regarding the concentrations of benzo[a]pyrene used in the study. However, the question regarding the discussion of previous work related to the role of oxidative stress in benzo[a]pyrene-induced neurotoxicity was not addressed. Since some aspects of the authors’ study were previously examined, the authors should address this in the discussion or at least explain why they don’t think it is necessary to add this information. Furthermore, the added text in the revised manuscript requires language correction.

Author Response

Response:

Thanks for the reviewer’s comment. I am very sorry that I misread and misunderstood the meaning of your first comment.

I have added the published information that related to the role of oxidative stress in benzo[a]pyrene-induced neurotoxicity to the discussion section (from line 612 to line 627) and the cited papers to the reference list.

Reviewer 3 Report

As requested, I reviewed the revised version of the manuscript (ID diagnostics-877402) "Integrated hypoxia signaling and oxidative stress in developmental neurotoxicity of Benzo[a]Pyrene in zebrafish embryos", by Yi-Chen Lin, Chang-Yi Wu, Chin-Hwa Hu, Tun-Wen Pai, Yet-Ran Chen, and Wen-Der Wang.

Please find below my considerations about authors’ responses.

I made the following minor comments:

-           In section 2.3 and on, authors should unify the indication of concentrations, i.e. avoiding to use 0% (percent) and 10 or 20 µM (micromolar)

THE REQUIRED CHANGES HAVE BEEN POSITIVELY ACCOMPLISHED.

-           Overall, many statements along the manuscript are in an almost hasty form, so they sound not entirely correct; please review the manuscript keeping in mind the explicit meaning of any statement.

AUTHORS DID NOT IMPROVE ACTUALLY THE ENGLISH FORM IN MANY STATEMENTS.

Major comments:

-           Lacking of sense probes as control for each antisense probe in situ assays is a serious shortcoming that needs to be amended

AUTHORS STATE TO HAVE ADDED THE IMAGES OF RESULTS WITH SENSE PROBES AS CONTROL (NAMELY IN SFIGURE 1).

-           In section 2.4 as well as in the related results, exposure time and embryonic stages must be clearly stated

THE REQUIRED CHANGES HAVE BEEN POSITIVELY ACCOMPLISHED.

-           In describing results of in situ hybridizations, authors usually give “quantitative” hints without the support of actual quantitative evaluation. For instance, regarding the elavl3 in situ assays, they state “dose-dependently reduced” without giving numerical demonstration. Also, for shh authors describe “obvious” differences: here, as in other points, they use the terms obvious or obviously in a criticizable manner, since nothing is obvious based only on the subjectivity of qualitative observation, especially when authors want to describe a scientific evidence. These issues concerning the interpretations of the in situ assays should be amended by performing staining quantitation for all the in situ images of all the gene-specific assays, just like the authors did for the assays of survivin and epo…

THE REQUIRED CHANGES HAVE BEEN POSITIVELY ACCOMPLISHED.

-           Section 3.4: if the authors want to compare effects of B[a]P with a positive control such as H2O2, they should demonstrate or, alternatively, describe and add comments on the potentially different routes of exposure between the two ROS-inducing situations on embryonic cells…

THE REQUIRED CHANGES HAVE BEEN POSITIVELY ACCOMPLISHED.

Author Response

Response 1:

Thanks for the reviewer’s comment.

I have send this manuscript to professional English Editing companies and the English form has been edited twice by native English speakers.